# Effectiveness and Safety of Dual Versus Triple Antibiotic Therapy for Treating Brucellosis Infection: A Retrospective Cohort Study

**DOI:** 10.3390/antibiotics14030265

**Published:** 2025-03-05

**Authors:** Yazed Saleh Alsowaida, Shuroug A. Alowais, Rema A. Aldugiem, Hussah N. Albahlal, Khalid Bin Saleh, Bader Alshoumr, Alia Alshammari, Kareemah Alshurtan, Thamer A. Almangour

**Affiliations:** 1Department of Clinical Pharmacy, College of Pharmacy, University of Hail, Ha’il 55473, Saudi Arabia; 2Department of Pharmacy Practice, Pharmacy of College, King Saud bin Abdulaziz University for Health Sciences, Riyadh 11461, Saudi Arabia; owaiss@ksau-hs.edu.sa (S.A.A.); aldugiem118@ksau-hs.edu.sa (R.A.A.); 421210040@ksau-hs.edu.sa (H.N.A.); salehk@ksau-hs.edu.sa (K.B.S.); 3King Abdullah International Medical Research Center, Riyadh 11481, Saudi Arabia; 4Pharmaceutical Care Department, King Abdulaziz Medical City, Riyadh 11481, Saudi Arabia; 5Department of Health Informatics, College of Public Health, University of Hail, Ha’il 55473, Saudi Arabia; b.alshoumr@uoh.edu.sa; 6Department of Pharmaceutics, College of Pharmacy, University of Hail, Ha’il 55473, Saudi Arabia; alia.alshammari@uoh.edu.sa; 7Department of Internal Medicine and Adult Critical Care, College of Medicine, University of Hail, Ha’il 55473, Saudi Arabia; k.alshurtan@uoh.edu.sa; 8Department of Clinical Pharmacy, College of Pharmacy, King Saud University, Riyadh 12372, Saudi Arabia; talmangour@ksu.edu.sa

**Keywords:** brucellosis, brucella, zoonotic infection, dual therapy, triple therapy

## Abstract

**Background:** Brucellosis is a major zoonotic infection that warrants treatment with antibiotic therapy. Current treatment recommendations include using either dual or triple therapy with antibiotics active against brucella species. This study aims to evaluate the effectiveness and safety of dual and triple antibiotic therapy for treating brucellosis. **Methods:** This is a retrospective cohort study for patients with confirmed Brucellosis infection from 2015 to 2024. The primary outcome was the achievement of a favorable response. Secondary outcomes were treatment failure, 90-day mortality, relapse of brucella infection, hospital re-admission, and adverse drug reactions (ADRs). Baseline characteristics were reported as means with standard deviations. All the statistical tests are shown as odds ratios (ORs) with 95% confidence intervals (CIs). **Results:** In total, 966 patients were screened and 287 met the inclusion criteria: 164 patients in the dual therapy group and 123 patients in the triple therapy group. Achievement of a favorable response was not statistically different between the dual therapy and triple therapy groups; 87.3% vs. 90.5%, OR 1.2 (0.48–3.02, *p* = 0.42). No patient died in either treatment group. Treatment failure, mean duration of hospitalization, brucella relapse, hospital re-admission, and the mean time to defervescence were not statistically different between dual and triple therapy groups. Adverse drug reactions were numerically higher in the triple therapy group. **Conclusions:** Dual therapy was equally effective for the treatment of patients with brucellosis compared to the triple therapy regimens. Although not statistically significant, there more ADRs in the triple therapy group compared to those receiving dual therapy. Thus, dual antibiotic therapy is efficacious, less costly, and associated with fewer ADRs compared to triple antibiotic therapy.

## 1. Introduction

Brucellosis is a zoonotic disease mainly affecting livestock and wildlife due to *Brucella* spp., a Gram-negative bacilli infection. This contagious illness has profound public health implications and poses an enormous economic risk, particularly in areas with inadequate food safety, hygiene, and veterinary care [1]. Brucellosis infection could be recurrent and often involves multiple organs and systems. Osteoarticular involvement is the most common complication, with a prevalence ranging from approximately 2% to 77%, and typically manifests as peripheral arthritis, sacroiliac arthritis, or spondylitis [1,2]. Hepatosplenomegaly is observed in approximately 50% of patients, while gastrointestinal symptoms such as nausea, vomiting, and abdominal pain are frequently reported. Although respiratory complications are rare, cases of pneumonia, pleurisy, pleural effusion, and pulmonary nodules have been documented. Furthermore, infections of the male genitourinary system, predominantly presenting as unilateral epididymal-orchitis or orchitis, occur in approximately 2% to 20% of cases. Additionally, endocarditis is responsible for more than 80% of deaths attributed to brucellosis [3]. The overall mortality rate of the disease is around 1%, while the incidence of brucellosis-associated endocarditis remains below 2%.

The most recent data published by the Saudi Arabian Ministry of Health indicates that the incidence rate of brucellosis in 2020 was 6.77 cases per 100,000 individuals [1,2]. The main method of transmission of brucellosis is through the consumption of unpasteurized dairy products [4,5,6,7]. Certain individuals are at high risk of brucellosis, including those who work closely with animals or have contact with infected animal materials. Essential preventative strategies against brucellosis include the pasteurization of milk and dairy products and adequate cooking of meat, with a temperature range of 63 °C to 74 °C.

Brucellosis can lead to significant illness and potentially fatal outcomes if left untreated, making prompt diagnosis and effective management critical [1,8,9]. The current treatment recommendations, based on the Centers for Disease Control and Prevention (CDC) and the World Health Organization (WHO), include the use of either dual or triple antibiotic therapy. A laboratory study for patients infected with brucellosis revealed significantly more patients in the triple therapy group had undetectable Brucella DNA (*p* = 0.02) [10]. Dual therapy has advantages compared to triple therapy, including a simpler antibiotics regimen, a lower rate of ADRs, and potentially lower cost [11]. However, patient preference plays a major role in the treatment of brucellosis, because the decision to initiate dual or triple therapy could have an impact on subsequent patient adherence with pharmacotherapy. Moreover, some antibiotic regimens necessitate parenteral antibiotic therapy with aminoglycoside (for example, gentamicin and streptomycin), which may be not preferred by some patients because it warrants either hospitalization or Outpatient Parenteral Antimicrobial Therapy. Lastly, triple antibiotic therapy carries a higher risk of contributing to the development of antibiotic-resistant bacteria [12].

Evidence is conflicted regarding the effectiveness of dual vs. triple antibiotic therapy for the treatment of brucellosis [13]. A study conducted in Saudi Arabia found that dual and triple therapy were equal in achieving a clinical cure rate [14]. The limitation of this study is that the sample size was small and may not be adequate to detect clinical differences. On the other hand, a meta-analysis by Skalsky et al. evaluating the treatment of human brucellosis from randomized controlled trials found that dual therapy was associated with a significantly higher rate of therapeutic failure compared to triple therapy [15]. Therefore, there is a need for studies with rigorous designs and adequate sample sizes to provide clinicians with robust evidence of whether or not dual and triple antibiotic therapy are equal in the treatment of brucellosis. Therefore, the objectives of this study are to evaluate the effectiveness and safety of dual and triple antibiotic therapy for the treatment of human brucellosis.

## 2. Results

Out of 966 patients screened, 287 patients were included in the study, with 472 patients excluded primarily due to receiving outpatient treatment (Figure 1). Patients were categorized into dual therapy (n = 164) and triple therapy (n = 123) groups.

### 2.1. Baseline Characteristics

Male patients accounted for the majority of the cohort (69.7%) with similar distribution across both groups (*p* = 0.85). The mean age was 44.9 years with no significant difference between the dual and triple therapy groups (44.4 vs. 45.5; *p* = 0.70). Significantly more patients in the dual therapy group were more likely to work with cattle compared with the triple therapy group (31.1% vs. 19.5%; *p* = 0.03). The most common source of infection was dairy consumption, accounting for 161 patients (56.1%), followed by animal contact (9.8%), while 34.1% had an unknown source of infection, with no significant difference between groups (*p* = 0.97). Most patients (273, 95.1%) were treated in the general medicine ward, with no significant difference in the treatment setting between groups (*p* = 0.67). Hypertension was the most common comorbid condition, presenting in 33.4% of patients, followed by diabetes mellitus in 33.1% of patients and dyslipidemia in 21.6% of patients. Baseline laboratory values were normal at baseline and comparable between the groups, except for two significant differences; the baseline serum creatinine levels were higher in the dual therapy group (1.069 mg/dL vs. 0.815 mg/dL; *p* = 0.03) and the erythrocyte sedimentation rate was higher in the triple therapy group (46.6 mm/h vs. 57.1 mm/h; *p* = 0.01). Complete baseline characteristics are available in Table 1.

### 2.2. Treatment Outcomes

The primary outcome, the achievement of a favorable response, was observed in 131 patients (87.3%) receiving dual therapy and 105 patients (90.5%) receiving triple therapy, with no significant difference between the groups (*p* = 0.42). The OR for achieving a favorable response with triple therapy compared to dual therapy was 1.2 (95% CI: 0.48–3.02), suggesting no significant advantage of triple therapy over dual therapy.

For secondary outcomes, treatment failure was observed in 30 patients (18.3%) in the dual therapy group and 29 patients (23.6%) in the triple therapy group, with no statistically significant difference between the two groups (*p* = 0.27). The mean duration of hospitalization was shorter in the dual therapy group compared to the triple therapy group; 10.74 days vs. 17.41 days, respectively, *p* = 0.12. However, the difference was not statistically significant. Furthermore, the relapse rate was numerically higher in the dual therapy group compared to the triple therapy group; 7 patients (4.3%) versus 3 patients (2.5%) respectively, *p* = 0.40. The rate of hospital re-admission was similar for dual and triple therapy groups; 1.9% of patients in the dual therapy group were re-admitted vs. 4.1% in the triple therapy group, *p* = 0.27. Time to defervescence was slightly shorter in the dual therapy group (67.16 vs. 72.70 days; *p* = 0.69), but this difference was not statistically significant. Overall follow-up rates were high and comparable between the groups. Lastly, regarding the ADRs, there was lower rate of ADRs in the dual therapy group; however, the difference was not statistically significant: 15.9% vs. 25.2% respectively, *p* = 0.05. Complete statistics for outcome results are available in Table 2.

### 2.3. Antibiotics Regimens Used

The antibiotic regimens used in the treatment of brucellosis varied significantly among patients. Dual therapy was administered to 164 patients, with the most common combinations being doxycycline and aminoglycoside, given to 85 patients (51.83%), followed by doxycycline and rifamycin, given to 48 patients (29.27%). For triple therapy, the most common combination regimen, used for 52 patients, was doxycycline, rifamycin, and aminoglycosides (21.14%), followed by doxycycline, ciprofloxacin, and aminoglycoside, given to 36 patients (29.2%). A complete list of antibiotic regimens used is available in Table 3.

## 3. Discussion

Brucellosis is a zoonotic infection associated with debilitating complications that impair quality of life [4,16]. The ideal antibiotic treatment strategy for brucellosis has not been defined. The present study evaluated brucellosis’s treatment and safety outcomes in Saudi Arabia. We found no difference in the rate of favorable outcome achievement between the dual and triple therapy groups. Additionally, we found no difference between the dual and triple therapy groups in 90-day mortality, relapse of brucella infection, hospital re-admission, antibiotic discontinuation due to ADRs, doctor’s office follow-up, mean duration of hospitalization, or mean time for brucella symptoms resolution.

There are several species of brucella, with 12 species that are known to be pathogenic and cause infection [17]. Genetic variants are identified as a contributing factor to the development of brucellosis infection [18]. It is important to note the vaginal and gut microbiota of infected animals also play a role in the infectivity of the animal. Furthermore, the host systemic immune response is key for controlling the brucella infection once ingested or inhaled [19]. Therefore, hosts with immunocompromising conditions may have an increased risk for brucella infections. From a microbiological view, brucella has several virulence factors, including lipopolysaccharide, T4SS secretion system, and BvrR/BvrS, which permit brucella to interact with the host and form brucella-containing Vacuole that allow the bacteria to multiply [20,21].

Our study revealed that dual therapy is as effective as triple therapy, since we found that the achievement rate of a favorable outcomes and the therapeutic failure rate were not statistically different. The achievement rate of favorable outcomes was similar in the dual therapy (87.3%) vs. triple therapy group (90.5%), with *p* = 0.42 in our study, consistent with a study by Al-Madfaa et al. conducted in Jeddah, Saudi Arabia [14]. That study involved 54 patients, and the authors reported no difference in the clinical cure rates for the dual therapy group (86.2%) vs. the triple therapy group (80%), *p* = 0.54. Aljuaid et al. conducted a descriptive study for patients with spinal brucellosis in Makkah, Saudi Arabia [22]. Their study included 35 patients, with 94.2% of the patients on triple therapy and reported an 84.8% clinical cure rate for patients treated with triple therapy of various regimens. Furthermore, another study by Yang et al. for brucellosis in hospitalized Chinese patients included 100 patients. The authors found that the triple therapy group achieved a statistically significantly higher response rate of 84% compared to the dual therapy group’s rate of 66%, *p* < 0.001 [23]. Differences in the patient population, sample size, and antibiotic regimen variations used in the dual and triple therapy groups could explain the differences compared to our findings. Silva et al. conducted a network meta-analysis to evaluate several regimens for the treatment of brucellosis that contradict our findings in achieving favorable outcomes [24]. The authors found that triple therapy with doxycycline + streptomycin + hydroxychloroquine lowered the failure rate compared to dual therapy with doxycycline + streptomycin, reporting a relative risk 0.08 (95% CI: 0.01–0.76). The study by Silva et al. should be interpreted with caution, because they compared these regimens artificially using a network meta-analysis method and included hydroxychloroquine, which is not a standard of care for brucella infection.

In our study, the rate of brucella relapse was not statistically different between groups, with 4.3% in dual therapy and 2.5% in triple therapy, *p* = 0.4, findings which are incongruent with a study by Hasanain et al. [25]. The authors conducted a randomized controlled trial for patients infected with brucella in Egypt and found dual therapy had a significantly higher relapse rate compared to triple therapy, 22.6% vs. 9.3%, *p* = 0.01. A likely explanation for the difference in the relapse rate is the smaller sample size of the study by Hasanain et al. (107 patients) compared to our study, the difference in the patient population, and patient perception of behavioral changes including milk pasteurization. To elaborate, a meta-analysis by Huang et al. evaluated triple versus dual therapy for the treatment of brucellosis and found that triple therapy had a significantly lower relapse rate compared to dual therapy, reporting a relative risk 0.29 (95% CI: 0.18–0.45) [11]. The difference in findings between the study by Huang et al. compared to our study could be explained by the exclusion criteria. The investigators excluded patients with serious complications, such as endocarditis or neurological diseases, which makes the patient population not truly representative of brucella patients in the real world.

In our study, the time to defervescence was not statistically different between dual and triple therapy groups, *p* = 0.69. Therefore, this suggests the equal effectiveness of dual and triple therapies. Other studies that compared the time to defervescence for different combinations of dual therapy found no difference in time to defervescence [24,26]. Notably, baseline serum creatinine was statistically significantly higher in the dual therapy group; however, this is clinically meaningless since it is still within the normal range. Similarly, the baseline erythrocyte sedimentation rate was statistically significantly higher in the triple therapy group; however, it is not clinically significant because both dual and triple therapy groups had elevated erythrocyte sedimentation rates.

In our study, we proposed to evaluate the 90-day mortality rate; however, we did not report any deaths in either study group. That is consistent with other studies which found very low mortality related to brucella infection [26]. Furthermore, in our study, we evaluated hospital re-admission for dual and triple therapy groups, and we found no statistical difference, *p* = 0.27 (Table 2), which suggests a relatively equal effectiveness of dual and triple therapy groups for the treatment of brucellosis. Notably, no published studies evaluated hospital re-admission for dual and triple therapy groups to compare against our findings. Lastly, doctor’s office follow-ups were similar for dual and triple therapy groups, *p* = 0.99.

In our study, there was a higher rate of ADRs in the triple therapy group compared to the dual therapy since the amount of ADRs was numerically higher in the triple therapy group: 25.2% vs. 15.9%, OR 1.79 (95% CI: 0.98–3.2, *p* = 0.05). Our findings were consistent with a study by Hasanain et al. that they found a higher ADR rate in the triple therapy group compared to the dual therapy group. However, this was not statistically significant: dual therapy 11.3% vs. 20.4% for the triple therapy group, *p* = 0.05 [25]. It is not surprising that triple therapy is complex and associated with a higher likelihood of drug-related toxicities. Furthermore, another study by Yang et al. found statistically similar ADR rates for dual therapy (26%) vs. triple therapy (18%) groups, *p* = 0.33, which is consistent with our findings [23]. A meta-analysis by Huang et al. also found a similar rate of ADR between dual and triple therapy groups, with a relative risk of 0.85 (95% CI: 0.67–1.06, P0.11) [11]. Lastly, in our study, we did not find a difference in the rate of brucella therapy discontinuation between the dual and triple therapy groups (*p* = 0.11) despite a numerically higher discontinuation rate in the triple therapy group (74.2%) compared to the dual therapy group (53.8%).

Our study evaluated several combinations of dual and triple therapy groups. In the dual therapy group, the most common combination was doxycycline + aminoglycosides (51.8%) and doxycycline + a rifamycin (29.3%), as presented in Table 3. In the triple therapy group, the most common regimens were doxycycline + ciprofloxacin + an aminoglycoside (29.2%) and doxycycline + a rifamycin + an aminoglycoside (21.4%). Other regimens in other studies provide alternative options in case of tolerability issues, drug availability, and pregnancy compatibility. Notably, our most common antibiotic for the dual and triple therapy groups is consistent with a study by Almadfaa et al.; they evaluated the use of doxycycline + rifampin for the dual therapy group and doxycycline + rifampin + an aminoglycoside for the triple therapy group [14]. The treatment recommendations for brucellosis based on the CDC for adults and children > 8 years old is a combination of doxycycline + rifampin [8]. Trimethoprim/sulfamethoxazole can be used if there are contraindications for tetracyclines. In complicated cases, an aminoglycoside can be added or substituted with rifampin. In contrast, the WHO recommends doxycycline + aminoglycoside [27], or, alternatively, doxycycline + rifampin. The concern with rifamycin antibiotics is that they are associated with developing antibiotic-resistant brucella strains and therapeutic failure; therefore, alternative antibiotics may be preferable if no contraindications exist. A study conducted in Egypt revealed that patients treated with doxycycline + rifampin had 59.3% recurrence of infection [11]. Thus, several antibiotics are active against brucella, and choice should be based on the risk of resistance strains, toxicity profile, and patient preference.

Our study shed light on the effectiveness and safety of brucella treatment in Saudi Arabia with the advantage of a large sample size of 287 patients. However, our study has limitations. It is a retrospective observational study, which may introduce the risk of missing or incomplete information. Due to many possible dual and triple therapy combinations, we could not evaluate specific regimens for treating brucellosis. We evaluated the overall presence of ADRs without looking at specific toxicities. A few patients in the dual and triple therapy groups did not know their exact antibiotic regimens. Lastly, we could not confirm the outcomes with a polymerase chain reaction or DNA test, instead relying on the physician’s assessment.

## 4. Materials and Methods

### 4.1. Study Design and Setting

This is a retrospective cohort study for patients with brucella infection admitted to medical wards from 1 June 2015 to 30 March 2024. The study was conducted at King Abdulaziz Medical City (KAMC-R), Riyadh, Saudi Arabia, from January to November 2024.

### 4.2. Eligibility Criteria

Hospitalized adult patients and child patients 8 years or older admitted to the medicine ward or intensive care unit with a confirmed diagnosis of brucella infection based on laboratory serologic tests were included in the study. Patients must have been treated with active brucella antibiotics using either dual or triple treatment regimens. Pregnant patients and those with no follow-up data were excluded. Notably, the dosing regimens of antibiotics was adjusted and monitored by the clinical pharmacist and the practicing physicians. The duration of therapy was based on the patient’s clinical response, with a minimum treatment duration of 4 weeks.

### 4.3. Laboratory Test

The diagnosis of brucellosis was confirmed through a positive serological titer of ≥1:160 and/or the isolation of brucella species from blood cultures.

### 4.4. Data Collection

We collected the relevant data on the encrypted database: Research Electronic Data Capture (REDCap^®^) hosted by the University of Ha’il, College of Pharmacy. For every patient, we collected the following data from the electronic health record: demographics, living situation, source of infection, comorbidities, laboratory values (serum creatinine, procalcitonin, C-reactive protein, erythrocyte sedimentation rate, white blood cell count, transaminase liver enzymes), symptoms, antibiotics regimens, and clinical and safety outcomes.

The data collection was performed by PharmD students who received training to collect the data accurately. The students were supervised by pharmacy faculty and infectious diseases pharmacy consultants.

### 4.5. Outcomes

The primary outcome was the achievement of a favorable response. Secondary outcomes were treatment failure, 90-day mortality, relapse of brucella infection, hospital re-admission, ADRs, antibiotic discontinuation, office follow-up, mean duration of therapy, and time for brucella symptoms resolution.

### 4.6. Definitions

Favorable response: achievement of either complete clinical cure, defined as resolution of brucella symptoms, resolution of fever within 72 h (fever defined as a temperature of 38 °C or 100.4 °F or greater), and the resolution of leukocytosis (defined as white blood count less than 12 × 10^9^/L); or partial clinical cure, defined as achieving 1 or 2 of the criteria for complete clinical cure.Acute kidney injury is defined as alanine aminotransferase (ALT) ≥ 3 times the upper limit of normal in the presence of hepatitis or ≥5 times in the absence of symptoms [28].Recurrence of brucella: re-infection with brucella species confirmed with positive blood culture or serology within 90 days.Time for brucella symptoms resolution: time from antibiotic initiation to brucella symptoms resolution.

### 4.7. Statistical Analysis

Descriptive statistics were used to analyze the data. Bivariate comparisons for baseline characteristics and outcomes of interest for patients treated with dual and triple therapies were performed. Specifically, nominal data were reported as counts with percentages and analyzed using either the χ^2^ test or the Fisher exact test. Continuous data were presented as mean ± standard deviation with an independent *t*-test.

Study outcomes were analyzed using multivariate logistic regression using an OR with a 95% CI. Unbalanced baseline characteristics will be adjusted for in the logistic regression between dual and triple therapy groups.

A statistically significant test was achieved if the *p*-value < 0.05. All the statistical analysis was performed using Stata, version 18 (Stata Corporation, College Station, TX, USA).

## 5. Conclusions

Brucellosis is a major zoonotic disease with morbidity and healthcare costs. Antibiotics are the mainstay for the treatment of brucellosis. Dual antibiotic therapy is equally effective as triple antibiotic therapy for the treatment of brucellosis. Specifically, achievement of a favorable response, treatment failure, mortality, hospital re-admission, the mean time to defervescence, and relapse of brucella infection were comparable between the dual and triple therapy antibiotics groups. There was the potential for a better safety profile associated with dual antibiotic therapy for brucella treatment; dual antibiotic therapy is as efficacious, less costly, and associated with fewer ADRs compared to triple antibiotic therapy. Public education efforts should increase to encourage the pasteurization of dairy products to prevent the spread of brucella. Future studies should evaluate specific dual and triple therapy regimens for the treatment of brucellosis (for example, doxycycline + rifampin, doxycycline + rifampin + aminoglycosides, and doxycycline + Trimethoprim/sulfamethoxazole). Additionally, future studies should evaluate the therapeutic success of rifampin-based regimens.

## Figures and Tables

**Figure 1 antibiotics-14-00265-f001:**
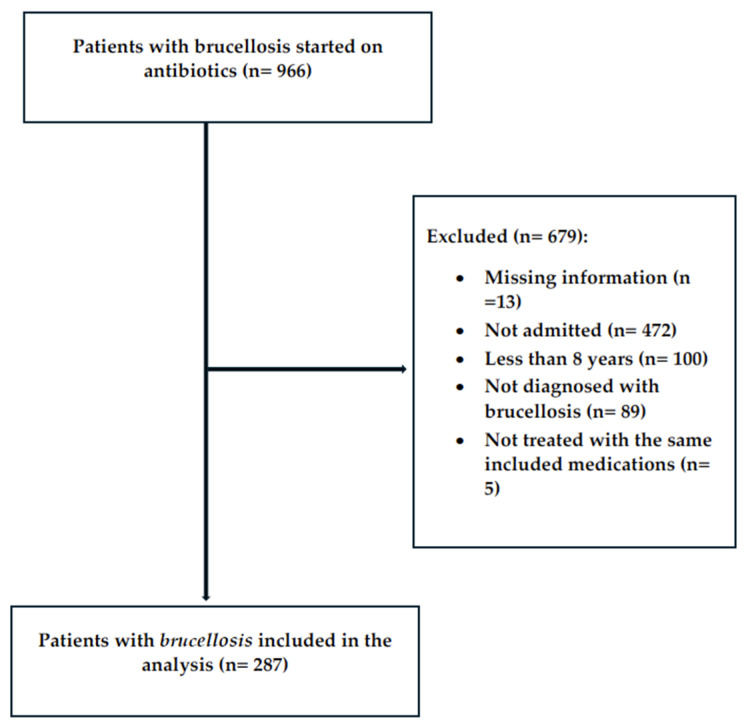
Patient screening diagram.

**Table 1 antibiotics-14-00265-t001:** Baseline characteristics.

Variables	Total N = 287	Dual Therapy N = 164	Triple Therapy N = 123	*p*-Value
Age, mean (SD)	44.9 (24)	44.4 (23.6)	45.5 (24.4)	0.70
Male gender	200 (69.7%)	115 (70.1%)	85 (69.1%)	0.85
Saudi national	282 (98.5%)	161 (98.7%)	121 (98.3%)	0.9
Weight, mean (SD)	67.5 (23.3)	68.3 (21.1)	66.6 (25.8)	0.56
Living situation Urban Rural Unknown	53 (18.5%) 12 (4.2%) 222 (77.4%)	23 (14.0%) 7 (4.3%) 134 (81.7%)	30 (24.4%) 5 (4.1%) 88 (71.5%)	0.08
Working with cattle	75 (26.1%)	51 (31.1%)	24 (19.5%)	0.03
Source of infection Dairy consumption Animal contact Unknown	161 (56.1%) 28 (9.8%) 98 (34.1%)	93 (56.7%) 16 (9.8%) 55 (33.5%)	68 (55.3%) 12 (9.8%) 43 (35.0%)	0.97
Treatment setting ICU General medicine	13 (4.5%) 274 (95.4%)	7 (4.3%) 157 (95.7%)	6 (4.9%) 117 (95.1%)	0.67
**Comorbidities**				
Hypertension	96 (33.4%)	52 (31.7%)	44 (35.8%)	0.47
Diabetes mellitus	95 (33.1%)	55 (33.5%)	40 (32.5%)	0.86
Dyslipidemia	62 (21.6%)	39 (23.8%)	23 (18.7%)	0.3
Chronic kidney disease	14 (4.9%)	11 (6.7%)	3 (2.4%)	0.1
Hypothyroidism	14 (4.9%)	9 (5.5%)	5 (4.1%)	0.58
Hyperthyroidism	1 (0.3%)	0 (0.0%)	1 (0.8%)	0.25
Malignancy	8 (2.8%)	7 (4.3%)	1 (0.8%)	0.08
Autoimmune disease	17 (5.9%)	14 (8.5%)	3 (2.4%)	0.03
Organ transplantation	3 (1.0%)	2 (1.2%)	1 (0.8%)	0.74
Taking immunosuppressive for cancer	7 (2.4%)	6 (3.7%)	1 (0.8%)	0.12
Cardiovascular disease	37 (12.9%)	21 (12.8%)	16 (13.0%)	0.96
**Symptoms**				
Myalgia	15 (5.2%)	9 (5.5%)	6 (4.9%)	0.82
Headache	42 (14.6%)	23 (14.0%)	19 (15.4%)	0.74
Arthralgia	27 (9.4%)	14 (8.5%)	13 (10.6%)	0.56
Fever	178 (62.0%)	123 (75.0%)	55 (44.7%)	<0.001
Sweating	71 (24.7%)	43 (26.2%)	28 (22.8%)	0.5
Low back pain	71 (24.7%)	33 (20.1%)	38 (30.9%)	0.04
**Focal disease**				
Peripheral arthritis	22 (7.7%)	9 (5.5%)	13 (10.6%)	0.11
Sacroiliitis	21 (7.3%)	6 (3.7%)	15 (12.2%)	0.006
Spondylitis	42 (14.6%)	13 (7.9%)	29 (23.6%)	<0.001
Epididymo-orchitis	9 (3.1%)	3 (1.8%)	6 (4.9%)	0.14
Baseline serum creatinine, mg/dL, mean (SD)	0.960 (0.81)	1.069 (1.2)	0.815 (0.42)	0.03
Acute kidney injury	29 (10.5%)	17 (10.8%)	12 (10.0%)	0.82
Baseline ALT	49.5 (SD 43.73)	50.4 (SD 44.4)	48.2 (42.8)	0.73
Baseline AST	52.4 (SD 51.3)	55.9 (58.4)	46.9 (37.6)	0.2
Liver injury	15 (5.2%)	8 (4.9%)	7 (5.7%)	0.76
Baseline C-reactive protein	51.3 (SD 47.8)	53 (SD 50.1)	46.1 (SD 40.3)	0.49
Baseline procalcitonin	0.65 (SD 2.3)	0.37 (0.66)	0.97 (3.34)	0.25
Baseline erythrocyte sedimentation rate, mm/h, mean (SD)	51.3 (SD 32.3)	46.6 (29.4)	57.1 (34.8)	0.01
Baseline white blood cell count	6.8 (3.4)	6.8 (3.9)	6.7 (2.4)	0.76
Baseline temperature	37.7 (0.98)	37.6 (0.96)	37.9 (0.98)	0.98
Mean duration of symptoms before therapy initiation	29.2 (75.3)	18.3 (23.6)	44.1 (111.4)	0.005
Previous brucella infection	44 (15.3%)	22 (13.4%)	22 (17.9%)	0.3
Co-infection	22 (7.7%)	13 (7.9%)	9 (7.3%)	0.85
Mean duration of therapy, week	8.2 (5.4)	6.987 (4.6)	9.854 (6)	<0.001
Positive culture for brucella	205 (71.7%)	119 (73.0%)	86 (69.9%)	0.57

**Table 2 antibiotics-14-00265-t002:** Outcomes.

Outcome	Dual Therapy N = 164 (%)	Triple Therapy N = 123 (%)	Odds Ratio (95% CI)	*p*-Value	Adjusted Odds Ratio (95% CI) *
Achievement of favorable response	131 (87.3%)	105 (90.5%)	1.38 (0.63–3.04)	0.42	1.2 (0.48–3.02)
Treatment failure	30 (18.3%)	29 (23.6%)	1.38 (0.78–2.45)	0.27	1.06 (0.52–2.16)
90-day mortality	0	0		-	
Mean duration of hospitalization, days (SD)	10.74 (1.7)	17.41(1.9)	-	0.12	-
Relapse of brucella after completion of therapy	7 (4.3%)	3 (2.5%)	0.56 (0.14–2.21)	0.4	0.43 (0.08–2.4)
Hospital re-admission	3 (1.9%)	5 (4.1%)	2.25 (0.53–9.6)	0.27	2.74 (0.45–16.55)
ADRs related to antibiotic therapy	26 (15.9%)	31 (25.2%)	1.79 (0.98–3.20)	0.05	1.73 (0.86–3.47)
Antibiotic discontinuation due to ADRs	14 (53.8%)	23 (74.2%)	2.46 (0.81–7.51)	0.11	2.83 (0.58–13.92)
Office follow up	151 (92.6%)	114 (92.7%)	1 (0.41–2.47)	0.99	0.52 (0.17–1.56)
Mean time to defervescence, days (SD)	67.16 (8.1)	72.70 (10.1)	-	0.69	-
Mean duration of therapy, weeks	6.987 (4.6)	9.854 (6)	-	<0.001	-

*: In the logistic regression, we adjusted for baseline serum creatinine, animal contact, presence of autoimmune disease, presence of spondylitis, mean duration of symptoms before therapy, and mean duration of therapy. Abbreviations: ADRs: adverse drug reactions, SD: standard deviation, CI: confidence interval.

**Table 3 antibiotics-14-00265-t003:** Antibiotics regimens used.

Dual Therapy, n = 164	Triple Therapy, n = 123
Doxycycline + aminoglycoside, n = 85 (51.83%)	Doxycycline + Rifamycin antibiotics + aminoglycosides, n = 52 (21.14%)
Doxycycline +rifamycin, n = 48 (29.27%)	Doxycycline + ciprofloxacin + aminoglycoside, n = 36 (29.2%)
Doxycycline + ciprofloxacin, n = 22 (13.4%)	Doxycycline + rifampin +ceftriaxone, n = 4 (3.2%) Ciprofloxacin + doxycycline + rifamycin, n = 7 (5.7%)
Tripmethoprim/sulfamethoxazole + Rifampin, n = 3 (1.8%)	Trimethoprim/sulfamethoxazole + rifampin+ gentamycin, n = 4 (3.2%)
Tripmethoprim/sulfamethoxazole + doxycycline, n = 2 (1.2%)	Trimethoprim/sulfamethoxazole + doxycycline+ aminoglycoside, n = 4 (3.2%)
Amoxicillin/clavulanates + doxycycline n = 1 (0.6%)	Trimethoprim/sulfamethoxazole +ciprofloxacin + doxycycline, n = 3 (2.4%)
Rifampin + cotrimoxazole, n = 1 (0.6%)	Ceftriaxone + rifamycin + doxycycline, n = 2 (1.6%)
	Cotrimoxazole + rifamycin + ceftriaxone, n = 1 (0.8%)
	Ampicillin + rifamycin + doxycycline, n = 1 (0.8%)
	Cefazolin + rifamycin + doxycycline, n = 1 (0.8%)
	Ciprofloxacin + rifamycin + aminoglycoside, n = 1 (0.8%)
	Piperacillin/tazobactam + ciprofloxacin + doxycycline, n = 1 (0.8%)
	Unknown triple therapy, n = 6 (4.8%)

## Data Availability

The data will be made available by the corresponding author upon request.

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
