# Peer review of "Effectiveness and Safety of Dual Versus Triple Antibiotic Therapy for Treating Brucellosis Infection: A Retrospective Cohort Study"

_antibiotics, 2025, doi:10.3390/antibiotics14030265_

Round 1
Reviewer 1 Report
Comments and Suggestions for Authors
The authors present a retrospective study on the effectiveness and safety of dual versus triple antibiotic therapy for treating Brucellosis infections in a large cohort over a 10-year period. The author’s study reviewed treatment options in 287 patients treated for Brucellosis with either a two-antibiotic or a three-antibiotic regimen. These findings suggest that dual therapy is as effective as triple therapy with less ADRs reported. This specific finding, three drugs versus two drugs having an increase in ADRs, is not unexpected anytime an increase in number of drugs used in treatments risk of adverse outcomes may occur.
Introduction – Introduction is clear and easy to follow the study with a straightforward introduction to the disease and the rates of infection and mortality. Clearly stated the benefits of dual vs triple regiments for treating Brucella infections. Also, present other studies with limited (n) however, the meta study conducted by Skalsky contradicts the author’s findings. The authors should present a clearer argument or more supportive analysis of their findings to support their conclusions other than Saudi data have small sample size.
line 44 – authors state “Brucellosis is a chronic and relapsing illness” – are authors referring to current infection after treatment or are the authors referring to untreated infections without mortality.
Line 71 – authors discuss DNA finding in triple therapy patients – at what stage of treatment? Does this statement translate to a positive outcome/effective treatment? Unclear the statement and the point. Needs clarification if the authors are using that study to support their conclusions.
Results – Figure 1 patents included n=288 in text 287.
Baseline characteristics and table 1 – in discussion section authors could note differences in serum creatine levels and erythrocyte sedimentation rates – as sedimentation rates are elevated in both data sets.
Treatment outcomes – authors use primary outcome term without defining for this study – is primary outcome the elimination of detectable infection by PCR/DNA analysis, time to defervescence, negative bacterial culture?
Define several terms in text and then use abbreviations – standards in writing – example
Adverse drug reaction (ADR) – abbreviated in text written out in table 2 – perhaps add under 4.6 additional medical terminology – this could expand the audience beyond clinicians. Another example – defervescence
Discussion – authors describe results from their study and discuss results from Yang which found effectiveness with a 3-antibiotic regiment. What differences in patient populations and sample size could account for the differing results in Yang versus the authors conclusions? These should be stated or made clearer and discussed. Not the same regimen at all?
Conclusions – authors could elaborate on why use rifampin regimen – most effective, less ADR, lease costly – unclear why authors are suggesting. Need clarification here.
One key piece of analysis that is missing from this study that would assist with advancing the treatment regimens for Brucella infections, especially since the authors concluded that the dual was as effective with lesser ADRs than triple antibiotic regimens should be a latitudinal study across dual treatments. This reviewer would suggest that the authors conduct an analysis to determine the effectiveness of variations of dual treatments. Is there significance in the outcomes using the dual combinations in this study? If the authors cannot include in this manuscript perhaps allude to the latitudinal study in their conclusions as a next step to further clarify best course of action for treating Brucella infections.
Author Response
Reviewer 1
Comment 1:
however, the meta study conducted by Skalsky contradicts the author’s findings. The authors should present a clearer argument or more supportive analysis of their findings to support their conclusions other than Saudi data have small sample size.
Response1: thank you for your comment, we revised the sentence to provide a clearer argument, and now the sentence reads as follows “Therefore, there is a need for studies with rigorous design and adequate sample size to provide clinicians with robust evidence of whether or not dual and triple antibiotic therapy are equal in the treatment of brucellosis.”
Comment 2:
line 44 – authors state “Brucellosis is a chronic and relapsing illness” – are authors referring to current infection after treatment or are the authors referring to untreated infections without mortality.
Response 2: thank you for your comment, we meant by that current infection after treatment, patients can have a relapse of the illness. To avoid confusion, we revised the sentence, and it now reads as “Brucellosis infection could be recurrent and often involves multiple organs and systems.”
Comment 3:
Line 71 – authors discuss DNA finding in triple therapy patients – at what stage of treatment? Does this statement translate to a positive outcome/effective treatment? Unclear the statement and the point. Needs clarification if the authors are using that study to support their conclusions.
Response 3: thank you for your comment, this point provide basis for our research since scientists in laboratory studies found that triple therapy is preferable because more patient had undetectable brucella DNA. However, clinical research is needed to evaluate the clinical outcome and effectiveness of dual vs triple therapy for treating brucellosis.
Comment 4: Results – Figure 1 patents included n=288 in text 287.
Response 4: thank you for your comment, there is a typo we corrected that in Figure 1.
Comment 5: Baseline characteristics and table 1 – in discussion section authors could note differences in serum creatine levels and erythrocyte sedimentation rates – as sedimentation rates are elevated in both data sets.
Response 5: thank you for your comment, baseline serum creatinine is slightly elevated in the dual therapy group; however, it is still within the normal range since it is <1.2 mg/dL and it is clinically meaningless. Seimilarly, the erhtythocyte sedemntation rate is statistically higher in the triple therapy group; hoever, it is clinically not significant. We added that in the discussion section to clarify that as follows” Notably, baseline serum creatinine was statistically significantly higher in the dual therapy group; however, it is clinically meaningless since it is still within the normal range. Similarly, the baseline erythrocyte sedimentation rate was statistically significantly higher in the triple therapy group; however, it is not different clinically because both dual and triple therapy groups had elevated erythrocyte sedimentation rates.”
Comment 6: Treatment outcomes – authors use primary outcome term without defining for this study – is primary outcome the elimination of detectable infection by PCR/DNA analysis, time to defervescence, negative bacterial culture?
Response 6: thank you for your comment, unfotunatly we relied on the physician assessment for our outcomes and we could not ascertain that with PCR/DNA test. We added that as a limitation in the discussion as follows “Lastly, we could not ascertain our outcomes with polymerase chain reaction or DNA test, and we relied on the physician’s assessment.”
Comment 7: Define several terms in text and then use abbreviations – standards in writing – example
Adverse drug reaction (ADR) – abbreviated in text written out in table 2 – perhaps add under 4.6 additional medical terminology – this could expand the audience beyond clinicians. Another example – defervescence
Response 7: we corrected all the typos and made uniform abbreviations throughout the manuscript.
Comment 8: Discussion – authors describe results from their study and discuss results from Yang which found effectiveness with a 3-antibiotic regiment. What differences in patient populations and sample size could account for the differing results in Yang versus the authors conclusions? These should be stated or made clearer and discussed. Not the same regimen at all?
Response 8: thank your comments, it is already mentioned in the same paragraph that the study by Yang et al. included hospitalized chinese patients and included 100 patients as a sample size as follows “ Furthermore, another study by Yang et al. for brucellosis in hospitalized Chinese patients included 100 patients.”
Regarding antibiotic regimen used, in our study, we used multiple dual and triple therapy regimens and our are not similar to the study of Yang et al. we highlighted that in the discussion section as follows “Differences in the patient population, sample size, and antibiotic regimen variations used in the dual and triple therapy groups could explain the difference compared to our findings.”
Comment 9: Conclusions – authors could elaborate on why use rifampin regimen – most effective, less ADR, lease costly – unclear why authors are suggesting. Need clarification here.
Response 9: thank you for your comment, to clarify, rifampin-based regimes are associated with therapeutic failure and resistance and are not preferable. We added a sentence to the conclusion section to highlight your point as follows “Thus, dual antibiotic therapy is efficacious, less costly, and associated with fewer ADRs compared to triple antibiotic therapy.”
Comment 10: One key piece of analysis that is missing from this study that would assist with advancing the treatment regimens for Brucella infections, especially since the authors concluded that the dual was as effective with lesser ADRs than triple antibiotic regimens should be a latitudinal study across dual treatments. This reviewer would suggest that the authors conduct an analysis to determine the effectiveness of variations of dual treatments. Is there significance in the outcomes using the dual combinations in this study? If the authors cannot include in this manuscript perhaps allude to the latitudinal study in their conclusions as a next step to further clarify best course of action for treating Brucella infections.
Response 10: thank you for your comment, unfortauntely, this not feasible to do in our study because we have varity of dual therapy regiemns and for some regimens there is only limited number of patients. For example, in the dual therapy of dxycycline + ciprofloxacin there is only 22 patients, so this would not provide meaningful findings. However, we added to the conclusoin section for the future studies to consider as follows “Future studies should evaluate specific dual therapy regimens for the treatment of brucellosis (for example, doxycycline + rifampin, doxycycline + aminoglycosides, and doxycycline + Trimethoprim/sulfamethoxazole.”

Reviewer 2 Report
Comments and Suggestions for Authors
- The authors didn’t specify the dose of each drug used, nor the duration of the therapy for both the dual and triple group, so it is not known whether the dual therapy cases was using higher dose and/or longer course. Please comment
- The authors claim that the exclusion of patients with serious complications is the reason that the study by Huang et al. has contradicting result to this study in the relapse rate. So, would similar result be observed if the authors apply the same exclusion criteria to this study? Please explore and comment.
- Can the authors comment on the potential effect of dual versus triple antibiotic treatment in controlling antibiotic resistance?
- I would recommend the authors to re-format/re-organize the tables for better readability.
Author Response
Effectiveness and safety of dual versus triple antibiotic therapy for treating Brucellosis infection: A retrospective cohort study
Dear Editor-in-chief,
We would like to thank you and the reviewers for carefully reviewing the manuscript and providing comments to improve its quality. We have addressed all the comments, as shown in the revised manuscript. Please note that all changes has been highlighted in the manuscript for convenience.
Reviewr 2
Comment 1: The authors didn’t specify the dose of each drug used, nor the duration of the therapy for both the dual and triple group, so it is not known whether the dual therapy cases was using higher dose and/or longer course. Please comment
Response 1: thank you for your comment, unfortunately, we did not collect the dosing information as well as the duration; however, the dose will be adjusted based on renal and hepatic function by the clinical pharmacist and the physician responsible for the patient. Additionally, the duration of therapy will be based on the patients clinical repones which vary from 6 weeks to 12 months. We added that to the method section for clarity for the readers as follows “Notably, the dosing regimens of antibiotics will be adjusted and monitored by the clinical pharmacist and the practicing physicians. The duration of therapy will be based on the patient’s clinical response with a minimum treatment duration of 4 weeks.”
Comment 2: The authors claim that the exclusion of patients with serious complications is the reason that the study by Huang et al. has contradicting result to this study in the relapse rate. So, would similar result be observed if the authors apply the same exclusion criteria to this study? Please explore and comment.
Response 2: thank you for your comment, we believe excluding patients with severe brucella complications (such as endocarditis and neurological complications) make the patient population healthier, and easier to treat. Brucellosis in real-life is more complicated and affect vital organs such as brain and heart with poor prognosis and treatment outcomes. This is why we believe the exclusion criteria by the study Huang et al. make triple therapy arm have less relapse. If included complicated brucella cases, they could have equal relapse rate compared to the dual therapy group.
Comment 3: Can the authors comment on the potential effect of dual versus triple antibiotic treatment in controlling antibiotic resistance?
Response 3: thank you for your comment, the more antibiotics prescribed, the more risk of antibiotic resistant bacteria. Therefore, the triple therapy carries a higher risk of antibiotic resistance, we added that to the introduction section to provide readers with that information as follows” Lastly, triple antibiotics therapy carry higher risk of development of antibiotic-resistant bacteria.”
Comment 4: I would recommend the authors to re-format/re-organize the tables for better readability.
Response 4: thank you for your comment, we re-formatted and re-organized several aspects of all tables in the manuscript.

Reviewer 3 Report
Comments and Suggestions for Authors
The authors presented the work "Effectiveness and safety of dual versus triple antibiotic therapy for treating Brucellosis infection: A retrospective cohort study" with aim of evaluating the effectiveness and safety of dual and triple antibiotic therapy for treating brucellosis.
The work is clear and relevant. It is well-written. The introduction provides the context and relevance. The methodological strategy is well documented, including details in all analysis steps.
Opportunities for improvement:
1. Legends of tables are so simple, mainly table 2. Please extend to provide a little more info.
2. Can you recognize the impact of a specific antibiotic (only 1) dominating the outcome? For example, the use of "Doxycycline + aminoglycoside" (an example) was always present in both main regimens with 2 or 3 antibiotics.
The dominant regimens were:
In 2 antibiotics: Doxycycline + aminoglycoside, n= 85 (51.83%)
With 3: Doxycycline + ciprofloxacin + aminoglycoside, n = 36 (29.2%)
In this case, if we compare those cases, is the p-value>0.05?
This will help to clarify the role of ciprofloxacin on the outcomes.
3. In the same line, in Table 3: The list of combinations is big. If you consider the class of antibiotic (having fewer categories), and compare the dominant regimens, do you think you can identify differences? Did you run an exploratory analysis?
4. Please, in the discussion, add a paragraph about the epidemiological triad, in which outcomes depend on the human (genetics, immunity, and morbidities), the pathogen (strain, genotype and virulence,) and the environment (exposition, use of drugs, etc). For example, different clinical profiles can be derived from an infection for the same pathogen, for example as in this study (please cite): https://pubmed.ncbi.nlm.nih.gov/35692458/
5. Conclusions can be extended to provide more details regarding the results based on outcomes, and also please add a sentence about the general approach.
Comments on the Quality of English LanguageThe text is well written.
Author Response
Reviewer 3
Comment 1: 1. Legends of tables are so simple, mainly table 2. Please extend to provide a little more info.
Response 1: thank you for your comment, we added the figure legend for the missing abbreviation. Other aspect of the table is self-explanatory and does not need legend explanations.
Comment 2: 2. Can you recognize the impact of a specific antibiotic (only 1) dominating the outcome? For example, the use of "Doxycycline + aminoglycoside" (an example) was always present in both main regimens with 2 or 3 antibiotics.
The dominant regimens were:
In 2 antibiotics: Doxycycline + aminoglycoside, n= 85 (51.83%)
With 3: Doxycycline + ciprofloxacin + aminoglycoside, n = 36 (29.2%)
In this case, if we compare those cases, is the p-value>0.05?
This will help to clarify the role of ciprofloxacin on the outcomes.
Response 2: thank you for your comment, unfortunately, we could not do subgroup analysis based on antibiotic regimen because the sample size will become small and unbalanced, so the interpretations will be inaccurate. We added that as a limitation in the discission section as follows “Due to many possible dual and triple therapy combinations, we could not evaluate specific regimens for treating brucellosis.” Additionally, we recommended for future studies, to evaluate specific antibiotic regimen in the conclusion section as follows” Future studies should evaluate specific dual and triple therapy regimens for the treatment of brucellosis (for example, doxycycline + rifampin, doxycycline + rifampin + aminoglycosides, and doxycycline + Trimethoprim/sulfamethoxazole.”
Comment 3: In the same line, in Table 3: The list of combinations is big. If you consider the class of antibiotic (having fewer categories), and compare the dominant regimens, do you think you can identify differences? Did you run an exploratory analysis?
Response 3: thank you for your comment, even if we consider the class, the number of combination will not change because majority of the combination have different antibiotic classes. Also, we could not do subgroup analysis based on antibiotic regimen because the sample size will become small and unbalanced, so the interpretations will be inaccurate.
Comment 4: 4. Please, in the discussion, add a paragraph about the epidemiological triad, in which outcomes depend on the human (genetics, immunity, and morbidities), the pathogen (strain, genotype and virulence,) and the environment (exposition, use of drugs, etc). For example, different clinical profiles
can be derived from an infection for the same pathogen, for example as in this study (please cite): https://pubmed.ncbi.nlm.nih.gov/35692458/
Response 4: thank you for your comment, we added the paragraph as recommended and cited the article you suggested as follows “There are several species of brucella, with 12 species that are known to be pathogenic and cause infection [16]. Genetic variants are identified as a contributing factor to the development of brucellosis infection [17]. It is important to note the vaginal and gut microbiota of the infected animals also play a role in the infectivity of the animal. Furthermore, the host systemic immune response is key for controlling the brucella infection once ingested or inhaled [18]. Therefore, hosts with immunocompromising conditions may have an increased risk for brucella infections. From a microbiological view, brucella has several virulence factors including lipopolysaccharide, T4SS secretion system, and BvrR/BvrS that permit brucella to interact with the host and form brucella containing Vacuole which allow brucella to multiply [19].”
Comment 5: 5. Conclusions can be extended to provide more details regarding the results based on outcomes, and also please add a sentence about the general approach.
Response 5: thank you for your comment, we provided more details in the conclusion based on our outcomes, and now the conclusion section reads as follows “Specifically, achievement of a favorable response, treatment failure, mortality, hospital re-admission, the mean time to defervescence, and relapse of brucella infection were comparable between dual and triple therapy antibiotics groups.”
